# Ethnicity and COVID-19 outcomes among healthcare workers in the UK: UK-REACH ethico-legal research, qualitative research on healthcare workers' experiences and stakeholder engagement protocol

Mayuri Gogoi ![ORCID],[1] Ruby Reed-Berendt,[2] Amani Al-Oraibi,[3] Osama Hassan,[3] Fatimah Wobi,[1] Amit Gupta,[4] Ibrahim Abubakar ![ORCID],[5] Edward Dove,[2] Laura B Nellums ![ORCID],[3] Manish Pareek ![ORCID],[1,6] UK-REACH Collaborative Group

MG and RR-B are joint first authors.

For numbered affiliations see end of article.

**Correspondence to**
Dr Manish Pareek;
mp426@le.ac.uk

## ABSTRACT

**Introduction** As the world continues to grapple with the COVID-19 pandemic, emerging evidence suggests that individuals from ethnic minority backgrounds may be disproportionately affected. The United Kingdom Research study into Ethnicity And COVID-19 outcomes in Healthcare workers (UK-REACH) project has been initiated to generate rapid evidence on whether and why ethnicity affects COVID-19 diagnosis and clinical outcomes in healthcare workers (HCWs) in the UK, through five interlinked work packages/work streams, three of which form the basis of this protocol. The ethico-legal work (Work Package 3) aims to understand and address legal, ethical and acceptability issues around big data research; the HCWs' experiences study (Work Package 4) explores their work and personal experiences, perceptions of risk, support and coping mechanisms; the stakeholder engagement work (Work Package 5) aims to provide feedback and support with the formulation and dissemination of the project recommendations.

**Methods and analysis** Work Package 3 has two different research strands: (A) desk-based doctrinal research; and (B) empirical qualitative research with key opinion leaders. For the empirical research, in-depth interviews will be conducted digitally and recorded with participants' permission. Recordings will be transcribed, coded and analysed using thematic analysis. In Work Package 4, online in-depth interviews and focus groups will be conducted with approximately 150 HCWs, from across the UK, and these will be recorded with participants' consent. The recordings will be transcribed and coded and data will be analysed using thematic analysis. Work Package 5 will achieve its objectives through regular group meetings and in-group discussions.

**Ethics and dissemination** Ethical approval has been received from the London-Brighton & Sussex Research Ethics Committee of the Health Research Authority (Ref No 20/HRA/4718). Results of the study will be published in open-access journals, and disseminated through conference presentations, project website, stakeholder organisations, media and scientific advisory groups.

## Strengths and limitations of this study

► Doctrinal and empirical research (Work Package 3) to understand ethical and legal implications in big data health research is novel to the UK Research Study into Ethnicity and COVID-19 Outcomes in Healthcare Workers (UK-REACH) study and has potential to inform policy and practice in the area.

► UK-REACH is the first comprehensive qualitative research (Work Package 4) with healthcare workers, particularly from ethnic minority backgrounds, in the UK exploring their experiences during the COVID-19 pandemic and perceptions of risk and coping mechanisms.

► The engagement of stakeholders (Work Package 5) in every stage of UK-REACH is exemplary and provides real-world relevance to the research and the findings.

► As the ethico-legal empirical research will recruit key opinion leaders (and not members of the wider population), the demographic diversity of the sample and the opinions gathered in interview may be limited by the characteristics of those in leadership positions in the field.

► Due to the pandemic restrictions, interviews and focus group discussions will be conducted via online methods as a substitute for face-to-face meetings, posing practical and technological challenges for dynamic interaction with participants.

**Trial registration number** ISRCTN11811602.

## INTRODUCTION

Since the start of the COVID-19 pandemic in December 2019 in China, the SARS-CoV-2 has spread rapidly to almost all parts of the world, infecting officially, and to date, more than 170 million people and claiming around 3.5

million lives.[1] As the world continues to fight against this novel virus, there is emerging evidence that ethnicity may be an important risk factor in COVID-19 infection, disease and mortality.[2–4] In the UK, people from ethnic minority communities have been found to be disproportionately affected by COVID-19.[5–9] Ethnic differences in COVID-19 outcomes have become significant because of the grave medical and clinical concerns, and due to accompanying issues of marginalisation and health inequities affecting these communities, which predate the pandemic.[10] Urgent calls had, therefore, been made to incorporate ethnicity into COVID-19 research, although with caution to adopt a holistic view of ethnicity.[11 12] While research on ethnicity and COVID-19 has since progressed,[13 14] its interplay with other crucial risk factors, such as occupation, remains scantily explored.

Occupational risk has been identified as a contributing factor in COVID-19 morbidity and mortality, with healthcare workers (HCWs) accounting for a large proportion of the total case load.[15] The increased burden of SARS-CoV-2 infection on HCWs and their families, particularly those from an ethnic minority background, has been reported, raising further concerns about the protection of those most at risk.[16–19] Potential explanations for increased risk among HCWs have mostly been attributed to patient-facing roles, lack of personal protective equipment, long working hours and even lack of training.[20] While these reasons may offer partial explanation, they fail to explicate the high rates of infection and deaths among ethnic minority HCWs. In their analysis of deaths from COVID-19 among National Health Service (NHS) workers in the UK, Cook *et al* point out that while ethnic minority workers constitute about 21% of the NHS workforce, they have accounted for nearly 63% of the total deaths.[19] These differential outcomes make an urgent case for exploring if, how and why ethnicity affects COVID-19 diagnosis and clinical outcomes in HCWs, with special reference to HCWs from ethnic minority groups.

The United Kingdom Research study into Ethnicity And COVID-19 outcomes in Healthcare Workers (UK-REACH), led by the University of Leicester, has been initiated to fill this gap. This study has been badged as an Urgent Public Health study by the National Institute for Health Research, and will run until August 2021 to rapidly examine differences in COVID-19 diagnosis, clinical outcomes, professional practices and physical and mental well-being among HCWs from different ethnicities through five interlinked work packages (see www.uk-reach.org for more details). Work Package 1 is an analysis of a linked data set with anonymised health data on COVID-19 outcomes, among clinical and ancillary HCWs [21]. Work Package 2 will establish a national longitudinal cohort of ethnic minority (with White ethnic group as comparator) HCWs and ancillary staff to assess changes in their health outcomes, social circumstances and professional roles over the course of the pandemic.[22] In this article, we describe the doctrinal, qualitative and stakeholder engagement protocol covering Work Packages 3–5 (WP3–WP5) of the project, which will explore questions of ethics, law, risk perception and behaviours of HCWs in relation to COVID-19.

## UK-REACH (WP3–WP5) objectives

The objectives of WP3–WP5 are to:

1. Undertake research to understand and address legal, ethical and social acceptability issues around data protection, privacy and information governance associated with the linkage of professionals' registration data and healthcare data (WP3).
2. Undertake qualitative interviews and focus groups with HCWs to examine experiences, risk perceptions, coping and support and physical and mental well-being pertaining to COVID-19 (WP4).
3. Develop a multiprofessional, national stakeholder group to inform the conduct of the research, and facilitate rapid dissemination and translation of the research findings into policy (WP5).

## METHODS
### Ethical and legal work package (WP3)
#### Study design

Research involving large data sets containing personal and health information raises legal, ethical and social issues surrounding the processing of such sensitive data, even when then the data are putatively anonymised. Understanding HCW concerns regarding trust, engagement, risk perception, barriers to participation and confidentiality is paramount for a project like UK-REACH to succeed and be conducted in a way that both respect participants' rights and interests, and also hold ethics and law at the forefront of each research activity. To do so, we will undertake two different strands of research. First, we will undertake desk-based doctrinal research to identify the relevant legal and ethical issues and provide a policy report with ongoing recommendations for implementation within the project. Second, we will conduct empirical qualitative research with key opinion leaders to explore their views on the ethical and legal implications of large data set analyses/cohort studies, such as risks of reidentification and identifying core principles of information governance in the context of sensitive data and HCW data sets. We explore each of these research strands in more detail below.

### Desk-based research

The ethico-legal work will commence with a comprehensive literature and doctrinal review based on consultation of relevant legal, regulatory and policy-based documents (conducted in part through consultation of the Westlaw legal database and the legislation.gov.uk website) to identify the legal framework and ethical issues pertinent to UK-REACH. This will focus on concerns surrounding privacy, data protection and human rights, and how to ensure an ethical approach for UK-REACH, particularly in the context of linking data concerning healthcare professionals' employment, registration and

health. Key issues we will consider include the limits of anonymisation, the risks of reidentification, particular ethical concerns arising from the use of sensitive data concerning ethnicity and what appropriate safeguards can be considered.

From this, a comprehensive policy report will be formulated to outline the key legal and ethical issues and provide recommendations for UK-REACH, to be delivered in month 3 of the project.

### Empirical qualitative research
#### Study population
The proposed empirical qualitative research will have key opinion leaders (≥16 years of age) working in a healthcare and biomedical research, or in health-related organisations (such as regulatory bodies, royal colleges, trade unions). Approximately 15–20 participants will be purposively selected and recruited through gatekeepers, members of the UK-REACH research team, stakeholder group and snowballing.

#### Data collection
Data will be collected using a semistructured topic guide, which has been developed by ED in consultation with other investigators and the Professional Expert Panel (PEP), which is the public involvement group in UK-REACH. The topic guide will include key areas of inquiry, such as participants' past experiences of research or knowledge of research processes, real or perceived barriers to research participation, views on current safeguards in law, policy and regulation around participants' rights and interests, exploring how ethnicity and race may influence risks and stigmatisation in research, or perceptions of the same and, finally, to gather views on protection measures that can or should be put in place by law or policy to adequately protect research participants. Additionally, for purposes of describing the cohort, basic demographic information such as age, gender, job role and geographic location will be collected using a short demographic data template.

Interviews will be conducted by a member of the research team via Microsoft Teams, or via telephone, depending on the availability, preference, COVID-19 limitations and/or work requirements of the participant. The interviews are likely to last between 45 and 60 min. Interviews will be recorded through the relevant platform software (eg, using the recording feature in Microsoft Teams) or on encrypted digital dictaphones, always with participants' express written permission. As interviews are conducted orally, data will be transcribed and then anonymised prior to data analysis.

#### Analysis plan
Digital files of the recorded interviews will be immediately uploaded securely and transcribed in intelligent verbatim by a transcription specialist company. The transcripts will be anonymised by removing all identifying information that enables indirect or inferential identification. Once

transcribed, we will compare the transcription with the recording to ensure accuracy.

The data from the interviews will be coded using qualitative thematic analysis. The process will consist of generating initial codes by comparing each of the transcripts. Coding is expected to be done manually and in multiple stages. We will adopt an inductive, data-driven approach and will begin with 'open coding', that is, reading each transcript (word by word and line by line). During the coding process, we will take notes in a memo-style format by writing down words and thoughts considered to be of use during the data analysis and serve as a reference for potential coding ideas. After completion of the open coding, initial codes will be constructed based on what emerged from the text, and we will proceed to code the remaining transcripts with those codes. When we encounter data that do not fit into an existing code, we will add new codes. We will then group the similar codes and place them into categories. These categories will be reorganised into broader, higher order categories, then grouped, revised and refined, and finally checked to determine whether the categories are mutually exclusive. At this point, we will form final categories, identifying subthemes both within and across the categories, which will then be organised into main themes.

### Qualitative research on HCW experiences work package (WP4)
#### Study design
We will undertake qualitative research with HCWs to understand their experiences during the COVID-19 pandemic. We will engage with clinical and ancillary staff from ethnic minority and White backgrounds working in healthcare settings (eg, front-line HCWs, ancillary staff working in hospitals, community practitioners) to gain insight into their perceptions around risk factors, support, coping mechanisms and their mental and physical health during the pandemic in order to inform response strategies to reduce COVID-19 morbidity and mortality in these individuals. We will conduct semistructured interviews and focus groups, which will enable in-depth explorations of individual participants' experiences and perspectives,[23] and also facilitate discussion between participants to explore both shared and differing experiences and perspectives.[24 25]

#### Study population
Participants will consist of adult individuals (≥16 years of age) with capacity to consent, from ethnic minority and White backgrounds with experience of working in healthcare settings during COVID-19, including both clinical and ancillary staff. We will recruit a purposive sample and will aim for theoretical saturation, including approximately 50 in-depth semistructured interviews and focus groups with a total of approximately 100 participants. Saturation describes the point at which no new data or insights are being gained from interviews or focus groups,[26] and so it becomes methodologically unnecessary to continue recruiting new individuals. In total, we aim to recruit

approximately 150 participants from different ethnicities, genders, job roles, hospital trusts and health boards, and UK regions to obtain a diverse sample.

## Data collection

We will recruit participants through collaborators/partners/stakeholders, community organisations and NHS organisations throughout the country. We will promote our research through posters, which will be advertised on-site and digitally (eg, online, by email and social media). Interested individuals will be able to contact the research team directly via the information provided on the poster. We will also work with gatekeepers in our partner and NHS organisations, who will send out communications regarding UK-REACH's qualitative research on HCW experiences to their networks or staff to facilitate recruitment. Additionally, a subset of cohort participants from the longitudinal cohort study (Work Package 2) who have given their consent to be contacted for further research will be invited to participate in the interviews/focus groups.

Semistructured interviews and focus groups will be conducted by the research team informed by a topic guide. The topic guide has been developed in consultation with the PEP members and piloted before commencement of actual data collection to trial out the questions as well as the online processes. Key topics included in this guide are: exploring participants' experiences of working during COVID-19; their fears and concerns at work and outside of work; perceived risk factors; challenges faced in accessing information to keep themselves safe; concerns around stigma, discrimination and racism; and identifying facilitators and coping mechanisms. To accommodate multiple participants, approximately 1.5 hours will be allocated to the focus groups compared with the 45–60 min for the one-to-one interviews. Following their participation, a token payment will be given to HCWs in recognition of their contribution to the research.

The topic guide will be the same when engaging healthcare staff through focus groups or one-to-one interviews. As in WP3, we will also collect basic demographic information about the participants using a short demographic data template. Interviews and focus groups will take place in a secure, virtual environment (eg, Microsoft Teams) or via telephone at a time that is convenient to the participants. Where prior consent is given, interviews and focus groups will be recorded through the relevant software platform (eg, using the recording feature in Microsoft Teams). Interview and focus group recordings will be transcribed by professional transcribers and pseudonymised before the start of analysis. The transcriptions will be supplemented with notes taken by the researchers during the interviews and focus groups.

## Analysis plan

Interview transcripts will be analysed using thematic analysis. Thematic analysis involves identifying themes or patterns in the data, lending coherence and order to it.[27] Following Braun and Clarke's six stages of thematic analysis, we will read and re-read the transcripts to build familiarity with the data, generate initial codes to develop a coding framework, collate codes into broad themes, review the themes, define and name the themes and, finally, write up the themes in a report form.[28] We will primarily adopt an open inductive approach to develop codes out of our data, but codes may also be developed from existing literature and/or previously conducted research.[27] Coding of data will be performed by the research team using NVivo software, and three/four different members of the research team will triangulate the coding process for credibility and rigour. Coding and theme development will be carried out until data saturation is reached and no new themes are emerging.

## Stakeholder engagement work package (WP5)

The rich diversity of the UK-REACH research will be complemented by a robust stakeholder involvement and engagement strategy, which has been in-built into the project (WP5) and conforms to the principles of (1) being receptive of public views and opinions, (2) collaborating and cocreating with the public, and (3) involving the public in wider dissemination of results. Within this work package, a stakeholder group (UK-REACH STAG) has been created to provide feedback and insights, and support in the formulation and propagation of the project recommendations. The group has membership from a range of partner stakeholder organisations (eg, General Medical Council, Nursing and Midwifery Council) and associations of ethnic minority professionals such as Filipino Nurses Association-UK and Association of Pakistani Physicians in Northern Europe. The group will meet virtually, once a month, until the end of the project, and will be governed by a set of terms of reference. Group meetings will be chaired by a nominated HCW chairperson/deputy chairperson, and a member of the research team will help with the coordination. Common email forums may also be created for the group members to share their views, opinions and feedback among each other outside of the periodic meetings. Progress with the delivery of other work packages, and where needed input from a stakeholder perspective is sought through these meetings. The stakeholder group's primary approach is through informal consensus building during the monthly meetings. Formal consensus approaches such as Delphi may be used if a more challenging decision need arises during the implementation of the project or for the purpose of optimising dissemination.

Views and opinions expressed by the group members will be aggregated, and individual names will not appear in any of the published documents. Meeting minutes will only be shared with the respective group members and on a need-to-know basis with members of the research team. The UK-REACH STAG will also provide support in the dissemination of the project recommendations.

## PATIENT AND PUBLIC INVOLVEMENT

Public involvement has been a central tenet in the UK-REACH project since its early stages. The project was developed in consultation and collaboration with national stakeholders including the General Medical Council, Nursing and Midwifery Council, royal colleges and ethnic minority HCW associations like British Association of Physicians of Indian Origin. The public involvement and engagement component has been further streamlined into the project with the creation of the PEP comprising healthcare professionals in various roles and from different ethnic backgrounds. The group members provide unique insight—relating to their professions and ethnic groups by virtue of their lived experiences—to certain aspects of the project. The PEP meets virtually and has provided inputs on the participant recruitment strategies for the different work packages, as well as questionnaire for the cohort study, topic guides for WP3 and WP4 and other study-related documents (eg, text within participant-facing items). We will continue to consult the PEP on other matters, such as data collection, analysis, reporting and even dissemination, as the project progresses.

## ETHICS AND DISSEMINATION
### Informed consent

Prior to focus groups and interviews, potential participants will be given participant information sheets (PIS), which will detail the nature of the research, objectives and any risks involved with participation. In light of COVID-19 constraints regarding face-to-face interaction, consent will be sought digitally (ie, via a secure internet portal) from the participants and a downloadable version of the completed form will be available to participants for their record. The right to decline to participate, and to withdraw consent at any stage of the research, will be explicitly stated on both the PIS and in discussion with potential participants. It will be explicitly stated that their signing of the consent form at no point supersedes their right to withdraw from the study. The PIS also states that if a participant withdraws from the study after collection of data, the collected data will be stored and analysed by the research team, unless the participant specifically requests for removal of the data at the time of withdrawal. The opportunity will also be given before every interview and focus group for participants to ask any questions about the scope of the research, or their rights as participants throughout the consent process.

### Psychologically or emotionally distressing conversations

While this study is low risk, particularly with respect to WP3–WP5, we recognise that exploring and discussing experiences around COVID-19 and ethnicity (including issues of stigmatisation, structural injustice or racism) could be distressing to participants. We aim to manage this risk through the consent process, clearly explaining to individuals what the study entails, and giving ample opportunities to question the process and decline to take part if individuals wish. We also aim to make the interview process as comfortable as possible, and ensure participants know they may stop, take a break or decide to withdraw from the interview and/or study at any point. The interview will always proceed at the comfort and discretion of the participant.

### Confidentiality and data protection

We will inform participants that participation will be confidential, and any personal information collected will be anonymised. Interview transcripts relating to individuals will also be pseudonymised using a unique numerical and date reference as the means to identify individual data sets. Such a system will ensure the anonymity of the participants and allow identification of individual data sets should a participant wish to exercise their individual rights (such as access, rectification or erasure). Individual data and transcripts will be held in secure digital drives, and original recordings will be deleted after transcription. Access to the full data set will only be provided to the members of the research team. The only circumstance in which individual-level data will be released is in the form of deidentified, anonymised excerpts within the final publication, which is a standard procedure in qualitative research of this type.[29] The excerpts will take the form of words, sentences and phrases the participants have provided which exemplify the coding framework and themes generated through the analysis.

### Dissemination of results and recommendations

We will ensure that the findings from the work packages are reported rapidly and published on our public-facing website (www.uk-reach.org). We have also enlisted the support of our stakeholders in disseminating the findings and recommendations through their organisational websites, newsletters, internal communications, blogs or social media channels like Twitter. Following suggestions from our STAG members, we will also endeavour to make recommendations available in other languages such as Welsh, for greater uptake. In addition, we will make our findings available to the Scientific Advisory Group for Emergencies and other policymakers in a timely manner so that policy decisions can be made in near real time. As a topic of immense public health significance, we will also endeavour to make our results available through print and electronic media. We will also publish the outputs of this research in peer-reviewed journals in line with the University of Leicester's Open Access publication policy to enable us to share the results widely with the academic community. We will also make presentations at relevant academic conferences as well as non-academic events organised by our collaborators.

## DISCUSSION

UK-REACH, led by the University of Leicester, is one of the first studies in the world that sets out to

understand why HCWs from minority ethnic backgrounds are disproportionately affected by COVID-19 as compared with their White counterparts. While emerging evidence from epidemiological studies is pointing to varied COVID-19 outcomes among different ethnic groups, not much is known about the reasons behind this variation. The qualitative work packages of the project, that is, WP3 and WP4, are expected to generate evidence which will be crucial in understanding some of the ethnicity-linked risk factors in COVID-19. The aim of the stakeholder engagement work package is to disseminate this evidence widely and in a timely manner. Additionally, our ethico-legal work package will generate important guidance on ways to minimise risks associated with research participation, and best practices in protecting the rights and interests of participants. Through our qualitative study on HCW experiences we aim to increase knowledge about risk perceptions, support and coping mechanisms relevant to COVID-19, which in turn will enable healthcare organisations to protect the mental and physical health of ethnic minority staff. We appreciate the insights that stakeholders can bring into our project and have enlisted their support from early on to maximise our reach and impact. It is the ultimate hope that through this project we will gain clear insight into the differences in COVID-19 clinical outcomes, professional practices and well-being among ethnic minority and White HCWs, in turn leading to a robust evidence basis for policymaking to minimise the impacts of COVID-19 on HCWs across the UK.

**Author affiliations**
[1]Respiratory Sciences, University of Leicester, Leicester, UK
[2]School of Law, The University of Edinburgh, Edinburgh, UK
[3]Division of Epidemiology and Public Health, School of Medicine, University of Nottingham, Nottingham, UK
[4]Oxford University Hospitals NHS Foundation Trust, Oxford, UK
[5]UCL Institute for Global Health, London, UK
[6]Department of Infection and HIV Medicine, University Hospitals of Leicester NHS Trust, Leicester, UK

**Collaborators** UK-REACH Collaborative Group members: Manish Pareek, University of Leicester, UK; Laura Gray, University of Leicester, UK; Laura B Nellums, University of Nottingham, UK; Anna Guyatt, University of Leicester, UK; Catherine Johns, University of Leicester, UK; Chris McManus, University College London, UK; Katherine Woolf, University College London, UK; Ibrahim Abubakar, University College London, UK; Amit Gupta, Oxford University Hospitals NHS Foundation Trust, UK; Keith Abrams, University of York, UK; Martin Tobin, University of Leicester, UK; Louise Wain, University of Leicester, UK; Sue Carr, University Hospital Leicester, UK; Edward Dove, University of Edinburgh, UK; Kamlesh Khunti, University of Leicester, UK; David Ford, Swansea University, UK; Rob Free, University of Leicester, UK.

**Contributors** MP and ED conceived and designed WP3. LBN, MP and AG conceptualised and designed WP4. IA and MP conceived and designed WP5. ED wrote the first draft of the WP3 protocol with inputs from RRB and MP. LBN wrote the first draft of the WP4 protocol with inputs from MP, AG, MG, AAO, OH and FW. MG wrote the first draft of the WP5 protocol with inputs from MP, IA and LBN. All authors were involved in writing, revising and approving the final manuscript. The UK-REACH Collaborative Group consists of all project collaborators who have conceptualised, designed and acquired ethical approval for the project. They have all read, refined and approved the final manuscript.

**Funding** UK-REACH is supported by a grant (MR/V027549/1) from the MRC-UK Research and Innovation and the Department of Health and Social Care through the National Institute for Health Research (NIHR) rapid response panel to tackle COVID-19. Core funding was also provided by NIHR Biomedical Research Centres. MP is funded by an NIHR Development and Skills Enhancement Award and also acknowledges support from the NIHR Leicester BRC and NIHR ARC East Midlands. LBN is supported by the Academy of Medical Sciences (SBF005/1047). This work is carried out with the support of BREATHE—the Health Data Research Hub for Respiratory Health (MC_PC_19004) funded through the UK Research and Innovation Industrial Strategy Challenge Fund and delivered through Health Data Research UK.

**Disclaimer** The views expressed in the publication are those of the author(s) and not necessarily those of the National Health Service (NHS), the NIHR or the Department of Health and Social Care.

**Competing interests** MP reports grants and personal fees from Gilead Sciences and personal fees from QIAGEN, outside the submitted work. IA reports personal fees from House of Lords, grants from Bill & Melinda Gates Foundation and grants from NIHR, outside the submitted work.

**Patient and public involvement** Patients and/or the public were involved in the design, or conduct, or reporting, or dissemination plans of this research. Refer to the Methods section for further details.

**Patient consent for publication** Not required.

**Provenance and peer review** Not commissioned; externally peer reviewed.

**ORCID iDs**
Mayuri Gogoi http://orcid.org/0000-0002-9946-2509
Ibrahim Abubakar http://orcid.org/0000-0002-0370-1430
Laura B Nellums http://orcid.org/0000-0002-2534-6951
Manish Pareek http://orcid.org/0000-0003-1521-9964

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
