## [Reviewer comments · BMJ Open]

ARTICLE DETAILS

TITLE (PROVISIONAL)	Ethnicity and COVID-19 outcomes among healthcare workers in the United Kingdom: UK-REACH ethico-legal research, qualitative research on healthcare workers' experiences, and stakeholder engagement protocol
AUTHORS	Gogoi, Mayuri; Reed-Berendt, Ruby; Al-Oraibi, Amani; Hassan, Osama; Wobi, Fatimah; Gupta, Amit; Abubakar, Ibrahim; Dove, Edward; Nellums, Laura; Pareek, Manish

VERSION 1 – REVIEW

REVIEWER	Golestaneh, Ladan Yeshiva University Albert Einstein College of Medicine, Medicine/Renal
REVIEW RETURNED	07-Mar-2021

GENERAL COMMENTS	In this article, Gogoi et al propose a broad research study that uses a multipronged approach to investigate the disproportionate impact of COVID19 on racial-ethnic minority health workers in the UK. This is a protocol paper that describes the planned study design. It is a comprehensive design that promises to deliver important information as to the drivers of race/ethnicity based outcomes inequities with COVID-19. The authors' access to a cohort of individuals with a single high risk occupation (eliminating the need to consider occupation as a confounding factor) and with a wide variety of racial/ethnic members is an advantage and promises to deliver. The authors describe 3 approaches to help elucidate underlying factors at play 1) An ethico-legal quandary about use of big data and any inferences derived2) A qualitative evaluation of individual experiences and notions with respect to the topic at hand3) Input from various stakeholders regarding inequities in outcomes and reasons for that. My suggestions follow: Abstract: For someone not well-versed in the work packages alluded to, an encompassing statement about what the authors are trying to achieve through the variety of methods described would be beneficial. i.e: are the authors doing a root cause analysis of inequities in COVID19 outcomes based on race/ethnicity using a rich database of health workers affected by the pandemic? And if so: are they then looking to develop policies aimed at closing structural/care gaps that they identify? How do the the three work-packages combine to achieve this goal? These areas are covered further along in the manuscript.
--

	The first bullet point on page 5 is confusing to me: are the authors proposing to elucidate any controversy around privacy or ethics that exists with respect to using UK-REACH data? Are they doing this to set the framework for how best to conduct the study so that it is acceptable to the subjects being studied? Please explain. Introduction: a link to the very useful website (www.uk-reach.org) can be provided in this section to lay the foundation for what follows. Page 6, line 28: the sentence reads as if those researchers that can most contribute to a better understanding of the inequities of COVID outcomes in minorities are members of the mainstream ethnicity (unclear who they are). Consider revising. Page 11 line 20: why not focus on recruiting stakeholders from racial/ethnic minority backgrounds? They have the life experience, can better understand the cultural nuances at play and can help to elucidate concerns that are not immediately forthcoming.
--	---

REVIEWER	Pecoraro, Valentina Nuovo Ospedale Civile Sant'Agostino Estense di Baggiovara
REVIEW RETURNED	29-Apr-2021

GENERAL COMMENTS	Authors submitted an interesting work and the protocol is well written. Racial disparities in COVID-19 outcomes may be partially attributed to higher comorbidity rates in certain ethnicity, but it is not clear if the ethnicity is an independent prognostic factor for COVID-19. I suggest to assess the differences in health outcomes considering the differential exposure to the virus, vulnerability to infection and on health, social and economic consequences of the disease
---

REVIEWER	Karani, George Cardiff Metropolitan University School of Health Sciences, Occupational & Environmental Public Health
REVIEW RETURNED	12-May-2021

GENERAL COMMENTS	The multi-disciplinary team should be commended for developing the protocol. While I agree that some of the issues raised in this review may have been covered in the main grant application, I would suggest to the team to consider the following with regard to clarifications for the rationale used:  1. The abstract should include some more details on how the three work packages are inter-linked and what is to be expected at the end of the work. Perhaps a schematic diagram included on linkages? 2. There are recent relevant references that should be included, and the information updated regarding the current number of COVID-19 cases and deaths. 3. I suggest consistency in the protocol in regard to the aims of individual work packages. Example, see work package 5 on pages 5 and 18; work package 4 pages 7 and 11 etc. 4. It is not clear from the protocol how the limitation listed on page 5 bullet point 4, '...is mitigated by ...wide variety of ethnic backgrounds..... work package 5.' 5. You state that 'ethnic minority workers constitute about 21% of the NHS workforce,' p7. Is the 'n' used in the work package 3 p9 and in the work package 4 p11 produce the outputs expected considering that you state on p12 that ' subjects recruited from different ethnicities, gendersa diverse sample?'
--

	In work package 4 when working with staff from white and ethnic minority backgrounds, is there any weighting used when recruiting subjects? 6. What is the bias, if any of recruiting subjects through 'gatekeepers,' p 9,p12? 7. There is a time frame indicated for desk-based research p9. Is it possible to provide a rough time frame for the other work packages? 8. How will confidentiality be addressed during focus group meetings, and when interpreters are required , e.g. p13 etc. These details are missing. 9. What happens when a subject withdraws from study after all data has been collected? 10. On page 16, some risks to the subjects are highlighted. Will you signpost them to where they can obtain support if needed? 11. Have you contacted BAME voluntary organizations, say in Wales , to collaborate in the project? 12. I suggest that the overall aim stated on p18, ' We aim..... on all HCWs in the UK,' is checked for accuracy.
--	---

REVIEWER	Simpson, Colin Victoria University of Wellington
REVIEW RETURNED	19-May-2021

GENERAL COMMENTS	This is a well written protocol describing workpackages 3-5 of the UK-REACH study. These workpackages include a doctrinal and empirical study on the use of personal data, a qualitative study of healthcare workers, and stakeholder engagement activity. The intention is for this project to provide an understanding of the ethical and legal implications of the REACH study, and perceptions about its use of data. This is important work, given the sensitivity of engaging with healthcare workers to fully understand the important issue, in particular, around stigma, discrimination and racism - identifying facilitators and coping mechanisms. All three workpackages methods are described in depth and seem appropriate. There is a limitation on page 5 - for work package 4, in particular, the use of online methods of interview needs to be described in more detail here. e.g. If the researchers are unfamiliar with these methods, a pilot is strongly recommended to understand the limitations of the software and the participants willingness to engage in the process. Participant information sheets may also differ from face to face, as the environments may be home etc (no alcohol etc.). The use of an interpreter may be a barrier if non-natural discourse is necessary (to ensure full and frank answers and discussion). Careful design and piloting is likely to ensure high quality evidence is produced and more detail needs to be provided in the protocol. The authors should describe in the main text how they will mitigate WP3 predominantly White background participants e.g. via purposeful sampling and describe in more detail how WP5 will support this process to ensure a wider sample of non-white are included. The consent form/PIS should be included in the submission (I didn't see this). Are WPs 3&4 standalone or will they be used to inform each other.
---

	Presumably there are also WPs 1&2? More background should be provided on the overall REACH programme of work.
--	---

VERSION 1 – AUTHOR RESPONSE

Reviewer: 1

Dr. Ladan Golestaneh, Yeshiva University Albert Einstein College of Medicine

In this article, Gogoi et al propose a broad research study that uses a multipronged approach to investigate the disproportionate impact of COVID19 on racial-ethnic minority health workers in the UK. This is a protocol paper that describes the planned study design. It is a comprehensive design that promises to deliver important information as to the drivers of race/ethnicity based outcomes inequities with COVID-19. The authors' access to a cohort of individuals with a single high risk occupation (eliminating the need to consider occupation as a confounding factor) and with a wide variety of racial/ethnic members is an advantage and promises to deliver.

The authors describe 3 approaches to help elucidate underlying factors at play

- 1) An ethico-legal quandary about use of big data and any inferences derived
- 2) A qualitative evaluation of individual experiences and notions with respect to the topic at hand
- 3) Input from various stakeholders regarding inequities in outcomes and reasons for that.

Reviewer 1 Comment 1

Abstract: For someone not well-versed in the work packages alluded to, an encompassing statement about what the authors are trying to achieve through the variety of methods described would be beneficial. i.e: are the authors doing a root cause analysis of inequities in COVID19 outcomes based on race/ethnicity using a rich database of health workers affected by the pandemic? And if so: are they then looking to develop policies aimed at closing structural/care gaps that they identify? How do the three work-packages combine to achieve this goal? These areas are covered further along in the manuscript.

Response: Thank you for the comment. We have added a sentence in the Introduction section of our Abstract which reflects how the three work packages contribute towards the broader aim of the UK-REACH study i.e understanding ethnic differentials in COVID-19 outcomes among HCWs in the UK. Please refer to Page 3 for revision.

Reviewer 1 Comment 2

The first bullet point on page 5 is confusing to me: are the authors proposing to elucidate any controversy around privacy or ethics that exists with respect to using UK-REACH data? Are they doing this to set the framework for how best to conduct the study so that it is acceptable to the subjects being studied? Please explain.

Response: We have revised this point and the section. Please refer to the relevant section on Page 5 for the revisions.

Reviewer 1 Comment 3

Introduction: a link to the very useful website (www.uk-reach.org) can be provided in this section to lay the foundation for what follows.

Response: Thank you for the suggestion. A reference to the project website, along with the link, has been made in the Introduction. Please see Page 7 for the inclusion.

Reviewer 1 Comment 4

Page 6, line 28: the sentence reads as if those researchers that can most contribute to a better

understanding of the inequities of COVID outcomes in minorities are members of the mainstream ethnicity (unclear who they are). Consider revising.

Response: We have revised this sentence and replaced the word 'mainstream' with 'incorporate' to avoid any confusion in understanding. Please see Page 6 for the change.

Reviewer 1 Comment 5

Page 11 line 20: why not focus on recruiting stakeholders from racial/ethnic minority backgrounds? They have the life experience, can better understand the cultural nuances at play and can help to elucidate concerns that are not immediately forthcoming.

Response: Thank you for this suggestion. We have made efforts to include stakeholders from ethnic minority professional groups/associations (e.g. British Association of Physicians of Indian Origin, Filipino Nurses Association UK –full list of partners are available in the website which has now been referenced in the Introduction section) and have a good representation of members from these groups/associations in our stakeholder group. We have revised the sentence on Page 15 to reflect the diversity in our stakeholder group.

Alongside the stakeholder group, we also have a Professional Expert Panel (PEP) which is comprised of healthcare workers (HCWs) from ethnic minority backgrounds who have been instrumental in informing the study processes through their own lived experiences (Please refer to Page 16 for details).

Reviewer: 2

Dr. Valentina Pecoraro, Nuovo Ospedale Civile Sant'Agostino Estense di Baggiovara

Reviewer 2 Comment 1:

Authors submitted an interesting work and the protocol is well written. Racial disparities in COVID-19 outcomes may be partially attributed to higher comorbidity rates in certain ethnicity, but it is not clear if the ethnicity is an independent prognostic factor for COVID-19. I suggest to assess the differences in health outcomes considering the differential exposure to the virus, vulnerability to infection and on health, social and economic consequences of the disease

Response: Thank you for the comment. We appreciate the relevance of the suggestion and we do explore people's risk of exposure in relation to their job roles, socio-economic vulnerabilities experienced as a result of working arrangements, and their personal and family lives. We are also exploring how issues of stigma, racism and discrimination may have influenced HCWs experiences and risk of exposure. While these topics are explored qualitatively within Work Package 4, the longitudinal cohort study (Work Package 2), is also examining these issues statistically in a bigger sample of health care workers. The protocol for Work Package 2 has been submitted to BMJ Open and is under review.

Reviewer: 3

Prof. George Karani, Cardiff Metropolitan University School of Health Sciences

The multi-disciplinary team should be commended for developing the protocol. While I agree that some of the issues raised in this review may have been covered in the main grant application, I would suggest to the team to consider the following with regard to clarifications for the rationale used:

Reviewer 3 Comment 1

The abstract should include some more details on how the three work packages are inter-linked and what is to be expected at the end of the work. Perhaps a schematic diagram included on linkages?

Response: Thank you for the suggestion. We have added a sentence in the Introduction section of our Abstract which reflects how the three work packages are interconnected and contribute towards the broader aim of the UK-REACH study i.e understanding ethnic differences in COVID-19 outcomes among HCWs in the UK. Please refer to Page 5 for revision.

Reviewer 3 Comment 2

There are recent relevant references that should be included, and the information updated regarding the current number of COVID-19 cases and deaths.

Response: Thank you for the comment. We have revised the figures and updated the references in the main text. The new citations have been added to the Reference section.

Reviewer 3 Comment 3

I suggest consistency in the protocol in regard to the aims of individual work packages. Example, see work package 5 on pages 5 and 18; work package 4 pages 7 and 11 etc.

Response: We have revised the protocol and addressed the inconsistencies throughout the manuscript.

Reviewer 3 Comment 4

It is not clear from the protocol how the limitation listed on page 5 bullet point 4, '...is mitigated by ...wide variety of ethnic backgrounds..... work package 5.'

Response: Thank you for this comment, we have now amended the limitation to better reflect the overall methods within WP3, which is that because we are recruiting key opinion leaders, this may limit the diversity of participants, both demographically and in terms of opinions gathered. To note, this is a limitation of the healthcare field more broadly and the lack of diversity in higher level roles, rather than a limitation of the research methods. Please refer to Page 5 for the revisions

Reviewer 3 Comment 5

You state that 'ethnic minority workers constitute about 21% of the NHS workforce,' p7. Is the 'n' used in the work package 3 p9 and in the work package 4 p11 produce the outputs expected considering that you state on p12 that ' subjects recruited from different ethnicities, gendersa diverse sample?' In work package 4 when working with staff from white and ethnic minority backgrounds, is there any weighting used when recruiting subjects?

Response: Thank you for this comment. We would wish to clarify that determination of sample sizes in Work Package 3 and Work Package 4 have been informed by the principle of data saturation and not by population proportionate sampling. As a study exploring experiences of ethnic minority healthcare workers, Work Package 4 has set out to recruit at least two-thirds of the sample from ethnic minority workers.

Reviewer 3 Comment 6

What is the bias, if any of recruiting subjects through 'gatekeepers,' p 9,p12?

Response: Thank you for the comment. We do acknowledge that purposeful sampling in qualitative research may create the scope for selection bias. While this cannot be avoided altogether, we tried to minimise this by applying a combination of recruitment strategies such as email invites through regulatory bodies and Trusts, and social media promotion in Work Package 4. In Work Package 3,

along with recruitment through gatekeepers, we have also applied snowball sampling as a strategy; this detail has been added to the protocol (Please see Page 9 for revision).

Reviewer 3 Comment 7

There is a time frame indicated for desk-based research p9. Is it possible to provide a rough time frame for the other work packages?

Response: UK-REACH has been funded as an Urgent Public Health study to August 2021, to provide expedited outputs that will be of direct relevance to the UK government. The cut-off date for all Work Packages will follow the project-timeline. The timeline information has now been included on Page 7 of the manuscript.

Reviewer 3 Comment 8

How will confidentiality be addressed during focus group meetings, and when interpreters are required, e.g. p13 etc. These details are missing.

Response: Issues of confidentiality in the online Focus Groups will be the same as in physical Focus Groups, and participants will be apprised of the group etiquettes and confidentiality statements before start of session. Recordings will be downloaded soon after completion of discussion, and deleted from the cloud storage to prevent downloading and sharing by any participant. We have removed the reference to interpreters as we have not had to use any so far and do not foresee using them going forward.

Reviewer 3 Comment 9

What happens when a subject withdraws from study after all data has been collected?

Response: If a participant withdraws from the study, unless they specify that they want their data to be removed, the collected data will be analysed. This is specified in the PIS and a line has been added in the protocol to clarify this. Please see Page 16 for revisions.

Reviewer 3 Comment 10

On page 16, some risks to the subjects are highlighted. Will you signpost them to where they can obtain support if needed?

Response: Yes, the research team has prepared a database of resources and participants will be signposted to these sources, if the risk is perceived to be significant by the researcher or participant.

Reviewer 3 Comment 11

Have you contacted BAME voluntary organizations, say in Wales, to collaborate in the project?

Response: Yes, we have representatives from several ethnic minority professional groups (with members from across the UK) in our stakeholder group.

Reviewer 3 Comment 12

I suggest that the overall aim stated on p18, 'We aim..... on all HCWs in the UK,' is checked for accuracy.

Response: We have revised the section mentioned in this comment and made it consistent with rest of the information in the protocol. Please refer to Page 18 for revision.

Reviewer: 4

Dr. Colin Simpson, Victoria University of Wellington

Reviewer 4 Comment 1

This is a well written protocol describing workpackages 3-5 of the UK-REACH study. These workpackages include a doctrinal and empirical study on the use of personal data, a qualitative study of healthcare workers, and stakeholder engagement activity. The intention is for this project to provide an understanding of the ethical and legal implications of the REACH study, and perceptions about its use of data. This is important work, given the sensitivity of engaging with healthcare workers to fully understand the important issue, in particular, around stigma, discrimination and racism - identifying facilitators and coping mechanisms.

Response: Thank you for the positive comment.

Reviewer 4 Comment 2

All three workpackages methods are described in depth and seem appropriate. There is a limitation on page 5 - for work package 4, in particular, the use of online methods of interview needs to be described in more detail here. e.g. If the researchers are unfamiliar with these methods, a pilot is strongly recommended to understand the limitations of the software and the participants willingness to engage in the process. Participant information sheets may also differ from face to face, as the environments may be home etc (no alcohol etc.). The use of an interpreter may be a barrier if non-natural discourse is necessary (to ensure full and frank answers and discussion). Careful design and piloting is likely to ensure high quality evidence is produced and more detail needs to be provided in the protocol.

Response: We appreciate this comment and we have included a line in our protocol to reflect the piloting we did to test out our online processes and topic guide (Please refer to Page 12 for inclusion). We have removed reference to interpreters as we have not had to use any so far and do not foresee using them going forward.

Reviewer 4 Comment 3

The authors should describe in the main text how they will mitigate WP3 predominantly White background participants e.g. via purposeful sampling and describe in more detail how WP5 will support this process to ensure a wider sample of non-white are included.

Response: Thank you for this suggestion. We have added a line on Page 9 to reflect the different sources and strategies of recruitment that will make a part of Work Package 3.

Reviewer 4 Comment 4

The consent form/PIS should be included in the submission (I didn't see this).

Response: The PIS and Consent Form can be made available at readers' request. Generic information about participation is available in our project website and the website link is provided in the protocol manuscript.

Reviewer 4 Comment 5

Are WPs 3&4 standalone or will they be used to inform each other
Presumably there are also WPs 1&2? More background should be provided on the overall REACH programme of work.

Response: Thank you for this comment. The UK-REACH project is designed in a manner that all the work packages are inter-linked and inform each other, however, at the same time being distinct in their use of methodology and approach. We have added a line in the Introduction section to give an overview of Work Packages 1 and 2. Please see Page 7 for addition.

VERSION 2 – REVIEW

REVIEWER	Golestaneh, Ladan Yeshiva University Albert Einstein College of Medicine, Medicine/Renal
REVIEW RETURNED	14-Jun-2021

GENERAL COMMENTS	The authors have responded to previous point raised by this reviewer.
---

REVIEWER	Pecoraro, Valentina Nuovo Ospedale Civile Sant'Agostino Estense di Baggiovara
REVIEW RETURNED	14-Jun-2021

GENERAL COMMENTS	The authors responded to the comments adequately
--